# Preparation and Photoelectric Properties of Pr-Doped p-Cu_2_O Thin Films Photocatalyst Based on Energy Band Structure Regulation

**DOI:** 10.3390/molecules28227560

**Published:** 2023-11-13

**Authors:** Yuchen Wei, Qinggong Ji, Kai Wang, Jian Zhang, Jinfen Niu, Xiaojiao Yu

**Affiliations:** 1School of Materials Science and Engineering, Xi’an University of Technology, Xi’an 710048, China; yweiaa@xaut.edu.cn; 2School of Science, Xi’an University of Technology, Xi’an 710048, China; jqg_work@163.com (Q.J.); kw225713@163.com (K.W.); zhangjian@xaut.edu.cn (J.Z.); niujinfen@xaut.edu.cn (J.N.)

**Keywords:** Pr-Cu_2_O, photocatalyst, energy band structure, photocatalytic degradation

## Abstract

A Pr-doped p-Cu_2_O thin film was prepared on indium tin oxide conductive glass by electrochemical deposition; the effect of Pr doping on the structure, morphology, and physicochemical properties of p-Cu_2_O was investigated. The results show that with the increase in Pr doping amount, the particle size of p-Cu_2_O increases, the absorption boundary redshifts, and the band-gap width decreases. Pr doping increases the flat band potential and carrier concentration of p-Cu_2_O; when the doping amount is 1.2 mM, the carrier concentration reaches 1.14 × 1024 cm^−3^. Compared with pure p-Cu_2_O, the charge transfer resistance of Pr-doped p-Cu_2_O decreases and the photocurrent and open circuit voltage increase, indicating that the carrier transfer rate is accelerated, and the separation efficiency of photogenerated electrons and holes is effectively improved. The result of a norfloxacin photocatalytic degradation experiment showed that the degradation rate of norfloxacin increased from 52.3% to 76.2% and Pr doping effectively improved the photocatalytic performance of p-Cu_2_O. The main reasons for enhancing the photocatalytic performance are that the band gap of Pr-doped p-Cu_2_O decreases, the Fermi level of Cu_2_O is closer to the valence band position, the hole concentration near the valence band, and the oxidation capacity increases, and more h^+^ oxidize norfloxacin molecules. In addition, the Pr in Pr-Cu_2_O acts as a conductor to guide electrons on the guide band to the crystal surface, which increases the contact between photogenerated electrons and dissolved oxygen, which is conducive to the formation of the active species ·O_2_^−^ and can effectively reduce the recombination of photogenerated carriers. In the process of photocatalytic degradation of norfloxacin, the main active species are ·O_2_^−^, ·OH, and h^+^, which play auxiliary roles. TOC tests show that the norfloxacin molecules can be effectively degraded into small molecule organic matter, CO_2_, and H_2_O in the presence of Pr-doped p-Cu_2_O.

## 1. Introduction

In recent years, with the extensive use of antibiotics, antibiotics continued to be detected in the water environment; the chemical properties of antibiotics are stable and are difficult to degrade naturally, seriously affecting human health [1]. Due to the low concentration of antibiotics in the water environment, the separation or complete degradation of antibiotics is a difficult problem in water treatment research [2]. Photocatalytic oxidation technology is a newly developed water treatment technology in recent years, which has the characteristics of low energy consumption, high efficiency, and no secondary pollution and is especially suitable for the treatment of low-concentration and difficult-to-degrade organic pollutants [3]. Photocatalysts are the core of photocatalytic oxidation technology, in which the semiconductor photocatalyst has become the focus of research in recent years because it can use sunlight as a driving force to degrade organic pollutants [4]. Therefore, it is crucial to explore an ideal photocatalyst with high light absorption, excellent photocatalytic performance, and outstanding stability [5] (e.g., BiOBr [6], WO_3_ [7], and BiPO_4_ [8]). As a typical p-type semiconductor, Cu_2_O is widely used for pollutant photodegradation, nitrogen fixation, and carbon dioxide reduction due to its suitable bandgap, strong response to visible light, low cost, and non-toxic properties [9,10,11].

However, the photogenerated electrons and holes produced by Cu_2_O under light are easy to recombine, which makes its quantum efficiency low [12]. Moreover, the oxidation–reduction potential of Cu^+^ lies between the Cu_2_O band gap, which makes it easy to photocorrode, thus limiting the application of Cu_2_O [13]. In order to improve the photocatalytic performance of Cu_2_O, ionic doping [14], metal modification [15], semiconductor composite [16], and other methods are mainly used to modify its structure. Studies have shown that the modification of Cu_2_O can effectively improve its photoelectric performance. Doping is widely used in the modification of Cu_2_O due to its simple preparation [17,18,19]. Kumar et al. [20] prepared Cu_2_O based nanocrystals doped with Na and Co by a sol-gel method. It was found that with the increase in the concentration of Na and Co doping, the agglomeration phenomenon of samples was gradually weakened. The band gap of Cu_2_O-based nanocrystals is reduced, and the optical absorption in the visible light region is improved. Ibrahim et al. [21] used radio frequency/direct current sputtering technology to deposit Ni-doped Cu_2_O thin films on glass substrates. The studies show that with the increase in Ni content, the grain size decreases gradually. When the doping amount is 2.6%, the photoband gap of Cu_2_O film decreases from 2.35 eV to 1.9 eV and the photocurrent density increases to −5.72 mA·cm^−2^, showing good photoelectrochemical properties and stability. Gu et al. [22] prepared sulfur-doped Cu_2_O and g-C_3_N_4_ composites by an in-situ synthesis method. The studies show that the photoelectron and hole recombination of Cu_2_O are inhibited by sulfur doping compared with the undoped precursor. The photocatalytic hydrogen production performance and stability of Cu_2_O are effectively improved. Ding et al. [23] synthesized B-doped CuO/Cu_2_O composite material and used it to degrade bisphenol A, and the kinetic degradation rate of bisphenol A was increased by 12 times. B doping promotes the Cu(II)/Cu(I) cycle and realizes the synergistic effect between free radicals and non-free radicals. It can be seen that metal or nonmetal doping is an effective way to improve the photoelectrochemical properties and stability of Cu_2_O.

In this paper, Pr-doped p-Cu_2_O (Pr-Cu_2_O) thin films were prepared by co-deposition to change the band structure of Cu_2_O, reduce the band gap, promote the separation of photogenerated electrons and holes, enhance the conductivity of Cu_2_O, make the photogenerated carrier transfer quickly and react, avoid the occurrence of photocorrosion, improve the stability of Cu_2_O, make the light absorption redshifted, and finally improve the utilization efficiency of visible light of Cu_2_O. In this paper, the effect of the Pr(NO_3_)_3_ doping amount on the film morphology and photoelectric properties was investigated. Norfloxacin (NOR) was used as the target degradation material to evaluate its photocatalytic performance. The research results can provide valuable photocatalysts for the treatment of wastewater containing antibiotics.

## 2. Results

### 2.1. Selection of Deposition Potential for Pr-Cu_2_O

In order to determine the deposition potential (Ea) range for the prepared Pr-Cu_2_O thin films, cyclic voltammetry tests were performed. Figure 1a shows the cyclic voltammetry curve of the electrodeposited p-Cu_2_O thin films. As Figure 1a shows, a reduction peak appears at −0.45 V, corresponding to the reduction of Cu^2+^ to Cu^+^. In the range of −0.2–0.6 V, the cathode deposition current first increases, and then decreases as the negative augmentation of the Ea. If the deposition potential is too large, Cu^+^ will be reduced to Cu. The experiment shows that the film will be irregularly agglomerated if the deposition rate is too fast. Figure 1b shows the XRD patterns of Pr-Cu_2_O thin films prepared under different deposition potentials. It can be seen from Figure 1b that the peak position of Cu_2_O is consistent with the standard card (JCPDS NO.05-0667) [24]. It is found that the (111) crystal face preferred orientation of Pr-Cu_2_O film is best when the deposition potential is −0.45 V. It has been reported that the photocatalytic performance and stability of Cu_2_O are positively correlated with the (111) plane diffraction peak intensity [25,26]. Therefore, the deposition potential of −0.45 V was selected for sample preparation.

### 2.2. Microstructure Analysis of Pr-Cu_2_O Thin Films

When the deposition potential is −0.45 V, the electrolyte pH is 11 and the electrolyte concentration is 0.03 M. The effect of the additional amount of Pr(NO_3_)_3_ on the crystal structure of p-Cu_2_O thin films is discussed, as shown in Figure 2. It can be seen from Figure 2 that the diffraction peak of the p-Cu_2_O thin film is consistent with the standard card (JCPDS NO.05-0667). When the amount of Pr(NO_3_)_3_ is 1.2 mM, the diffraction peak intensity of the sample’s (111) crystal plane is the largest, indicating that the sample M_2_ should have good photocatalytic performance [19,26].

As seen in Figure 3, XPS analysis is used to examine the sample’s chemical valence and composition, while EDS is used to analyze the content of various elements in the sample. Figure 3a shows the fitting curve for O 1s, and the peak at the binding energy of 531.4 eV is related to the defect of oxygen vacancy in the crystal [27]. The peak at the binding energy 532.3 eV corresponds to the adsorbed oxygen or hydroxyl oxygen on the surface of Cu_2_O, which may be caused by the adsorption of water vapor in the air. The peak at 530.3 eV corresponds to the binding energy of the Cu_2_O lattice oxygen [25,28]. It can be seen from Figure 3b that two peaks appear at the binding energy of 932.4 eV and 952.2 eV, corresponding to the binding energy of Cu 2p3/2 and Cu 2p1/2 of Cu_2_O, respectively. This indicates that the Cu in Cu_2_O exists in the prepared sample in the form of Cu^+^ [25,29]. In addition, because the binding energy of Pr 3d is too close to the binding energy of the Cu 2p orbital and the content is low, it is not detected (Figure 3c). EDS analysis of Pr-Cu_2_O confirmed the presence of the Pr element in the M_2_ sample, but its content was only 0.11% (Figure 3d), consistent with XPS analysis.

SEM analysis was used to further investigate the effect of the Pr doping amount on the morphology of p-Cu_2_O films and the results are shown in Figure 4. It can be seen from Figure 4 that the doping amount of Pr has a great influence on the morphology of Cu_2_O films. As shown in Figure 4a, when the doping amount is 0.9 mM, the grain of the film presents a standard triangular prism morphology, and the grain size is relatively uniform. As shown in Figure 4b, when the doping amount is 1.2 mM, the grain of the film begins to change, basically maintaining the triangular prism type but the crystal edges become rounded. As shown in Figure 4c, when the doping amount is 1.5 mM, the film completely loses the triangular prism structure, becomes irregular, and has many particles on the surface. This may be due to the large amount of Pr doping, resulting in Cu_2_O crystal defects. Figure 4d shows the SEM image of a pure Cu_2_O thin film, which is a standard triangular prism junction. In addition, as the Pr doping amount increases from 0.9 mM to 1.5 mM, the grain size of the sample gradually decreases, but it is larger than that of the undoped Pr sample. This is because Pr doping can hinder the aggregation of Cu_2_O, increasing grain size. As the concentration of Pr increases, the agglomeration phenomenon gradually increases, which may be due to their extremely small size and high surface energy during the crystallization and drying processes, resulting in a gradual decrease in size [30].

### 2.3. Photoelectric Properties Analysis of the Prepared Samples

Figure 5 shows the UV-Vis DRS diagram of the Pr-Cu_2_O thin films, and the band gap width can be calculated according to the literature [31,32]. Figure 5a shows the absorption boundary of the sample is about 650 nm, and the absorption intensity of Pr-Cu_2_O thin film changes compared with pure Cu_2_O. As can be seen from Figure 5b, the band gap obtained by Pr-Cu_2_O films also changes. The band gap widths of samples M_0_, M_1_, M_2_, and M_3_ are 1.98, 1.97, 1.94, and 2.00 eV, respectively. Sample M_2_ has the largest light absorption intensity and the smallest band gap width. The narrower the band gap, the more easily the sample can be excited by sunlight to produce photogenerated electrons and holes, which helps to improve its photocatalytic performance.

In order to investigate the conductivity type and carrier concentration of Pr-Cu_2_O thin films, the Mott–Schottky test was performed on the samples. The test results are shown in Figure 6. As can be seen from Figure 6d, the pure Cu_2_O film slope of the Mott–Schottky curve is less than zero, and it is a p-type semiconductor [33], which is hole conduction. It can be seen from Figure 6a–c that the conductive type of the Cu_2_O does not change after Pr doping. The carrier concentration and band potential of the sample are calculated from the slope of the Mott–Schottky curve. The carrier concentrations of samples M_0_, M_1_, M_2_, and M_3_ were 1.19 × 10^21^, 3.51 × 10^22^, 1.14 × 10^24^, and 1.75 × 10^20^ cm^−3^, and the flat-band potentials were 0.79, 0.97, 0.90, and 0.84 V, respectively. Compared with pure Cu_2_O thin films, the carrier concentration of Cu_2_O thin films doped by Pr is increased to a certain extent. When the doping amount is 1.2 mM, the carrier concentration reaches a maximum of 1.14 × 10^24^ cm^−3^. After Pr doping, the flat band potential of Cu_2_O increases and the Fermi level (*E_f_*) of Cu_2_O is closer to the valence band position (*E_VB_*), which increases the hole concentration near the valence band and increases the oxidation capacity. Some studies have pointed out that higher carrier concentration means a better photocatalytic effect [34] and sample M_2_ should have better photocatalytic performance.

Figure 7a shows the current–time curve of the prepared samples. It can be seen from Figure 7a that the photocurrent densities of samples M_0_, M_1_, M_2_, and M_3_ are 0.001, 0.036, 0.050, and 0.029 mA·cm^−2^, respectively. The photogenerated current density of sample M_2_ is significantly higher than other samples, reaching 0.050 mA·cm^−2^. Higher photocurrent means better photogenerated charge separation efficiency [35]. At the same time, the photocurrent curve of the sample has a slow downward trend, indicating that its structure is stable and not prone to photocorrosion [36].

Figure 7b shows the open circuit voltage of the prepared samples. As can be seen from Figure 7b, the open circuit voltage of the sample changes periodically and is relatively stable with the alternating of open and closed light. A stable photovoltage indicates that the Pr-Cu_2_O thin film has a long carrier lifetime. The open circuit voltages of M_0_, M_1_, M_2_, and M_3_ are 3.243, 3.243, 5.438, and 4.194 mV, respectively. Compared with the pure Cu_2_O thin film, the peak open circuit voltage of the Pr-Cu_2_O thin film is significantly increased. The results show that the doped films have obvious advantages, among which the open circuit voltage of sample M_2_ is the largest, indicating that sample M_2_ has a higher photoelectron-hole concentration and can realize more efficiently sunlight photocatalysis [37].

Figure 8 shows the AC impedance spectrum curve of Pr-Cu_2_O thin films. At high frequencies, the AC impedance curve shows a semicircle, which is a characteristic of the charge transfer process. The diameter of the semicircle is equal to the charge transfer resistance (Rct) at the interface. It can be seen from Figure 8 that the Rct values of Pr-Cu_2_O films are all smaller than the impedance of Cu_2_O films. The results show that the electron migration ability in Pr-Cu_2_O thin films is obviously enhanced. The impedance value of the sample M_2_ is the smallest, the migration and reaction ability of photogenerated electrons and holes is the strongest, and the separation efficiency of photogenerated electrons and holes is the largest [38,39].

### 2.4. Analysis of Photocatalytic Properties of Pr-Cu_2_O Thin Films

The photocatalytic performance of the prepared samples was evaluated by using NOR as the target degradation material. It can be seen from Figure 9a that the UV absorption of NOR during degradation changes significantly, and the characteristic peak value at 270 nm gradually decreases with the extension of light time, indicating that NOR can be degraded. It can be seen from Figure 9b that compared with pure Cu_2_O, Pr-Cu_2_O doped films have a certain degree of improvement in the NOR degradation rate. In the presence of the photocatalysts M_0_, M_1_, M_2_, and M_3_, the degradation rates of NOR are 53.0%, 65.0%, 77.5%, and 63.3%, respectively. The degradation rate of NOR is the highest when the doping amount is 1.2 mM. The doped film has a 24.5% increase, indicating that Pr doping can effectively improve the photocatalytic performance of Cu_2_O, which is consistent with the above performance analysis.

### 2.5. Analysis of Factors for Enhancing Photocatalytic Performance of Pr-Cu_2_O

Based on the above performance test and analysis, the band gap of Pr-Cu_2_O thin films is reduced, which is more conducive to absorbing visible light, generating photoelectron-hole pairs, and improving its photocatalytic performance. The increase in photocurrent, open circuit voltage, and carrier concentration indicates that the recombination probability of photogenerated electron-hole pairs decreases effectively after Cu_2_O is doped. The decrease in AC impedance indicates that the carrier migration rate increases. The defects in the crystal, as a good electron trap, can capture photogenerated electrons and reduce the photogenerated electron-hole recombination probability, which indicates that the photoelectric performance of p-Cu_2_O can be effectively improved by Pr doping.

In order to study and analyze the mechanism of photocatalytic enhancement of Pr-Cu_2_O thin films from the perspective of energy band structure, the conduction and valence band positions of the Pr-Cu_2_O thin films were calculated by Equations (1) and (2) [40,41,42].
*E_VB_* = *χ* − *E_e_* + 0.5*E_g_*(1)
*E_CB_* = *χ* − *E_e_* − 0.5*E_g_*(2)
*χ* = [(A)^a^(B)^b^]^1/a+b^(3)

The *χ* value was estimated using the electronegativity (A and B) of the semiconductor constituent atoms, as shown in Equation (3) (a and b indicate the number of atoms A and B in the semiconductor) [43]. *E_e_* is the energy of the free electron at the hydrogen standard potential, which is 4.5 eV. The conduction band and valence band positions of Pr-Cu_2_O films are −0.14 eV and 1.80 eV. The Fermi level is calculated by Equations (4) and (5) [44].
*E_fb_* (*in V* vs. *NHE*) = −*E_f_* (*in eV* vs. *vacuum*) + *V_H_* (*in V*) − 4.5(4)
*V_H_* (*in V*) = 0.059 (*pH_pzzp_* − *pH*)(5)

*V_H_* is the potential drop of the entire Helmholtz layer and *pH_pzzp_* is the point of zero potential. The Fermi level of Pr-Cu_2_O is 1.08 eV. By comparison, it can be found that after doping with Pr, the Fermi level of Cu_2_O changes significantly. The *E_f_* of p-Cu_2_O is closer to *E_VB_*, which increases the hole concentration near VB, enables more h^+^ to oxidize NOR molecules, or reacts with H_2_O to form active species ·OH. In addition, Pr in Pr-Cu_2_O acts as a conductor to guide electrons on the guide band to the crystal surface, which increases the contact between photogenerated electrons and dissolved oxygen, which is conducive to the formation of the active species ·O_2_^−^ and effectively reduces the recombination of photogenerated carriers. Therefore, the photocatalytic performance of Cu_2_O is significantly improved, which is consistent with the results obtained from the previous photoelectric performance tests. The mechanism of the photocatalytic degradation of NOR by Pr-Cu_2_O thin films is shown in Figure 10.

### 2.6. Photocatalytic Mechanism Analysis of Pr-Cu_2_O

The active species capture experiment was used to investigate the degradation mechanism of NOR by Pr-Cu_2_O thin films. The experimental results are shown in Figure 11a. The trapping agents are methanol (MT), isopropyl alcohol (IPA), and p-benzoquinone (BQ), which can capture h^+^, ·OH and ·O_2_^−^ in the solution, respectively. It can be seen from Figure 11a that when MT, IPA, and BQ were added to the degradation system, the degradation rates of NOR were 63.8%, 56.3%, and 22.1%, respectively, indicating that ·O_2_^−^ played a major role in NOR photocatalytic degradation, while ·OH and h^+^ played auxiliary roles. According to the capture experiment results and related literature [11,45,46], a possible catalytic mechanism was proposed, as shown in Equations (6)–(11).
Pr-Cu_2_O + hv (λ > 400 nm) → Pr-Cu_2_O (h^+^ + e^−^)(6)
Pr-Cu_2_O (e^−^) + O_2_ → ·O_2_^−^(7)
·O_2_^−^ + e^−^ + 2H^+^ → H_2_O_2_(8)
H_2_O_2_ + e^−^ + H^+^ → H_2_O + ·OH(9)
Pr-Cu_2_O (h^+^) + H_2_O → H^+^ + ·OH(10)
Active species (·OH, ·O_2_^−^, h^+^) + NOR → Degradation products(11)

Pr-Cu_2_O thin films can produce photogenerated electrons and holes under visible light (λ > 400 nm). Pr-Cu_2_O absorbs energy from the photon, the electrons in the valence band of Pr-Cu_2_O transition to the conduction band, and the holes remain in the valence band. Some of the photogenerated electrons are transferred to the surface of Pr-Cu_2_O, and ·OH is formed by a series of reactions with O_2_ adsorbed on the catalyst surface. The h^+^ can directly oxidize NOR molecules, and it can also react with H_2_O to form ·OH. Since ·O_2_^−^ plays a major role in the photocatalytic degradation of NOR molecules, a large number of active components of ·O_2_^−^ are produced in the reaction process and can contact NOR molecules adsorbed on the surface of Pr-Cu_2_O, and the NOR molecules can be degraded. Figure 11b shows the change in total organic carbon (TOC) during the degradation of NOR molecules. As can be seen from Figure 11b, TOC did not change much in the degradation system at the initial stage of degradation, indicating that NOR molecules degraded into small molecules of organic matter. TOC decreased rapidly in the degradation system at the intermediate stage, indicating that the degradation intermediates of small molecules were decomposed into CO_2_ and H_2_O. TOC did not change significantly in the final stage. It shows that some intermediate products are difficult to be completely decomposed. The change in TOC can also indicate that NOR molecules can be effectively degraded into small molecules of organic matter, CO_2_, and H_2_O in the presence of Pr-Cu_2_O.

## 3. Experimental

### 3.1. Preparation of Pr-Cu_2_O Thin Films

Pr-Cu_2_O thin films were prepared by the three-electrode electrochemical system at constant potential method. Using an indium tin oxide conductive glass (ITO, square resistance ≤ 15 Ω/sq) as the working electrode, a Pt electrode as the opposite electrode, a saturated calomel electrode (SCE) as the reference electrode, 0.03 M CuSO_4_ was mixed with 0.4 M lactic acid, a certain amount of praseodymium nitrate (Pr(NO_3_)_3_) solution was added, and the pH of the electrolyte was adjusted to 11 by NaOH. And the electrodeposition time was 60 min. The samples were taken out, washed in deionized water, and dried naturally. When the amount of Pr(NO_3_)_3_ added to the electrolyte was 0 mM, 0.9 mM, 1.2 mM, and 1.5 mM, respectively, the prepared samples were recorded as M_0_, M_1_, M_2_, and M_3_. The reagents used above were analytically pure and provided by Tianjin Kemiou Chemical Reagent Co., Ltd. (Tianjin, China).

### 3.2. Characterization and Photoelectrochemical Properties Test of the Samples

Using a 6100 X-ray diffractometer (Cu Target Kα Radiation, λ = 1.54056 Å, Shimadzu Instrument Co., Ltd., Kyoto, Japan) and K-Alpha X-ray photoelectron spectroscopy (XPS, Thermo Fischel Technology Co., Ltd., Waltham, MA, USA), the C 1 s transition at 284.6 eV was used as an internal reference to adjust the binding energy scale of peaks obtained for the experiments. A scanning electron microscope (VEGA3, Tesken Co., Ltd., Náchodská, Czech Republic) equipped with energy-dispersive X-ray spectroscopy (EDS) and UV-visible spectrophotometer (UV-3600, Shimadzu, Kyoto, Japan) were used to characterize and analyze the morphology, structure, elemental composition, and optical properties of the prepared samples. The TOC of the solution after degradation was tested using a TOC detector (vario TOC, Elementar, Langenselbold, Germany). An ultraviolet-visible spectrophotometer (UV-Vis, UV-3200PCS, Meipuda Instrument Co., Ltd., Shanghai, China) was used to monitor the concentration changes in NOR. A CHI660D electrochemical workstation (Shanghai Chenhua Instrument Co., Ltd., Shanghai, China) was used to test the electrochemical properties of the films.

### 3.3. Photocatalytic Degradation of Norfloxacin

The sample was photocatalyzed in the photocatalytic reaction instrument. Norfloxacin was used as the target degradation material. The sample (400 mg·L^−1^) was put into a quartz tube containing 50 mL of a 20 mg·L^−1^ NOR solution, the light source was a 250 W xenon lamp, and the adsorption balance was reached at 20 min. The light source was turned on for the photocatalytic reaction. The degradation system temperature was 35 °C. The degradation time was 3 h. The concentration of the NOR degradation solution was measured by the UV-Vis spectrophotometer at intervals of 30 min, and the degradation rate of NOR was calculated by conventional method.

## 4. Conclusions

In this paper, Pr-doped p-Cu_2_O thin film samples were prepared with different amounts of doped Pr(NO_3_)_3_. XRD, XPS, SEM, and UV-Vis DRS were used to characterize and analyze the structure and photoelectric properties of Pr-Cu_2_O thin films. NOR was used as the degradation target product to evaluate the photocatalytic properties of the thin film. The following conclusions can be drawn:

The prepared films have a preferred orientation of the (111) crystal face, and the band gap decreases from 1.98 eV to 1.94 eV. The narrower band gap is more susceptible to photoexcitation and transition. Pr doping significantly changes the morphology of p-Cu_2_O films and there are oxygen defects on the surface. Compared with pure Cu_2_O, the Pr-Cu_2_O thin film has higher photocurrent density, open circuit voltage, and lower impedance. When the doping amount is 1.2 mM, the prepared sample has good electron-hole separation efficiency and higher electron transfer rate and has better photocatalytic performance. The degradation rate of NOR is 24.5% higher than that of pure p-Cu_2_O.

The main reason for the improvement in the photocatalytic performance of p-Cu_2_O is the p-Cu_2_O reduced band gap after being doped with Pr. The Fermi level of the Pr-Cu_2_O is closer to the valence band, the hole concentration near the valence band is increased, the oxidation capacity is increased, and more norfloxacin molecules are oxidized by h^+^. At the same time, the Pr in the Pr-Cu_2_O acts as a conductor to guide the electrons on the guide band to the crystal surface, which is conducive to the formation of the active species ·O_2_^−^ and effectively reduces the photogenerated electron-hole pair recombination. The main active substance in the process of NOR photocatalytic degradation is ·O_2_^−^. In the presence of the Pr-Cu_2_O photocatalyst, NOR molecules can be effectively degraded into small molecule organic matter, CO_2_, and H_2_O. The results can provide an effective photocatalyst for the photocatalytic oxidation treatment of antibiotics in a water environment.

## Figures and Tables

**Figure 1 molecules-28-07560-f001:**
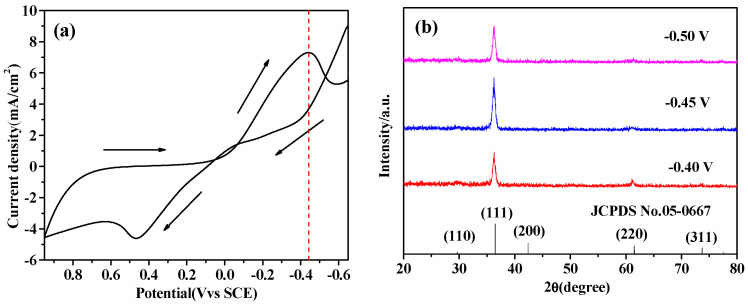
Cyclic voltammetry curve (**a**) and XRD patterns (**b**) of p-Cu_2_O.

**Figure 2 molecules-28-07560-f002:**
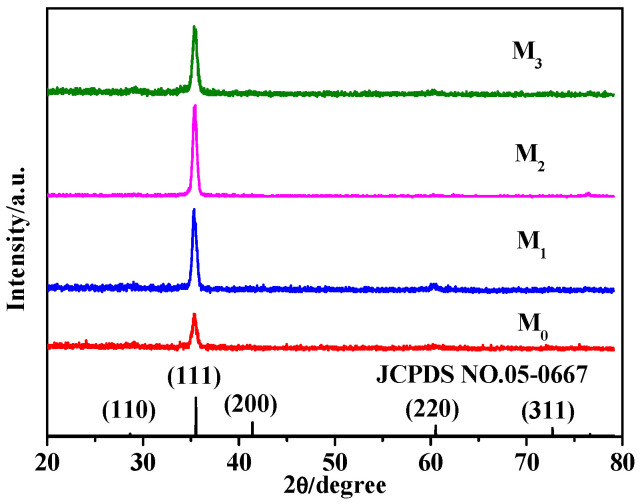
XRD patterns of the prepared samples.

**Figure 3 molecules-28-07560-f003:**
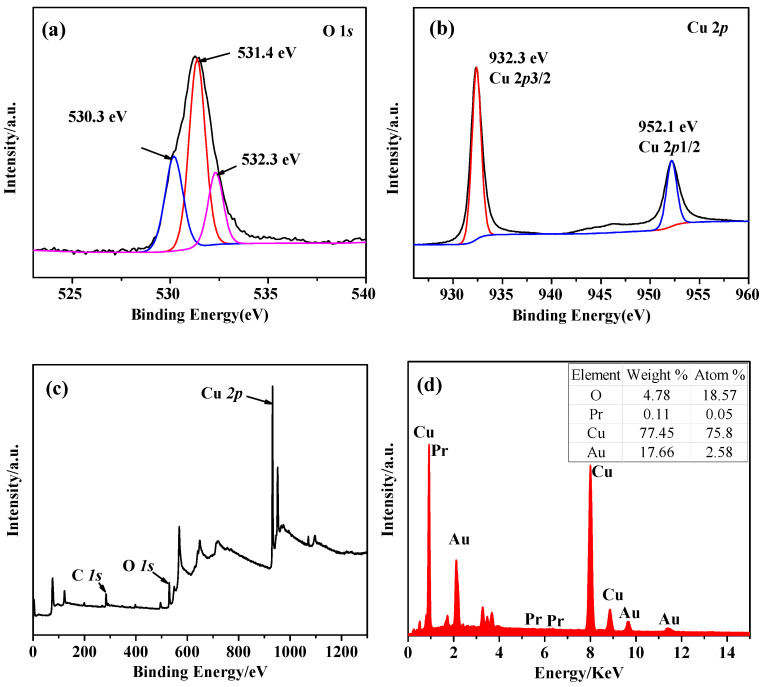
XPS pattern of Pr-Cu_2_O thin film, O 1s (**a**), Cu 2p (**b**), survey (**c**), and EDS (**d**).

**Figure 4 molecules-28-07560-f004:**
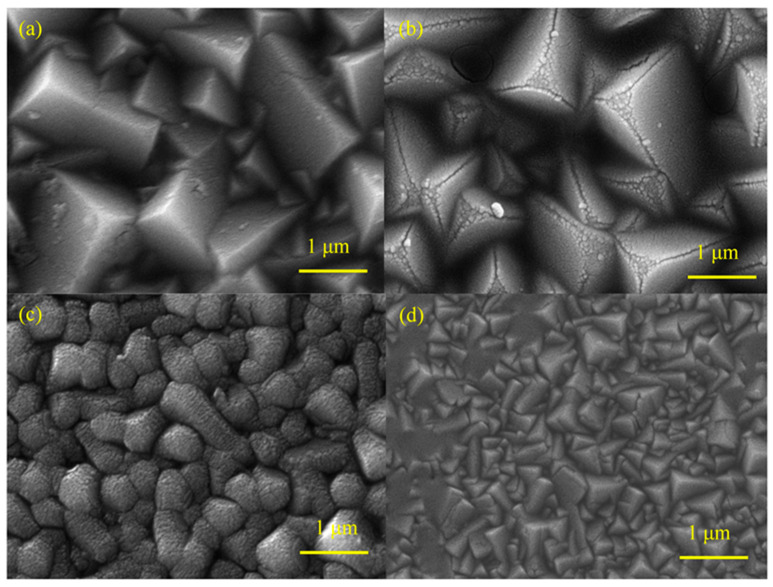
SEM images of the prepared samples M_1_ (**a**), M_2_ (**b**), M_3_ (**c**), and M_0_ (**d**).

**Figure 5 molecules-28-07560-f005:**
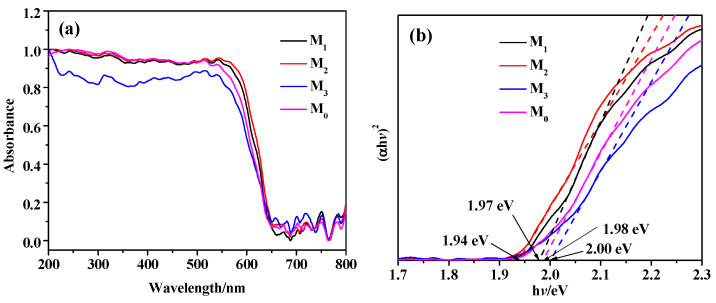
UV-Vis DRS (**a**) and (αhv)^2^-hv (**b**) spectra of the prepared samples.

**Figure 6 molecules-28-07560-f006:**
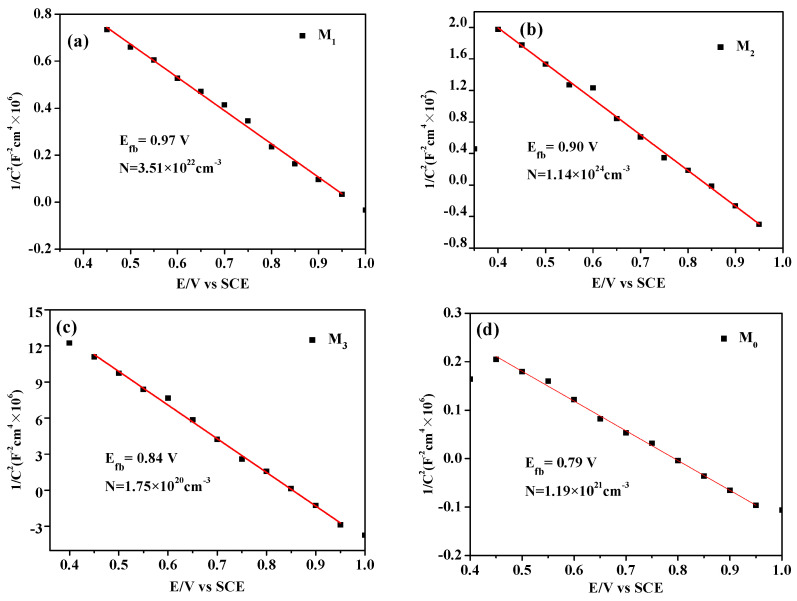
Mott–Schottky curves of the prepared samples under different doping amounts, 0.9 mM (**a**), 1.2 mM (**b**), 1.5 mM (**c**), and 0.0 mM (**d**).

**Figure 7 molecules-28-07560-f007:**
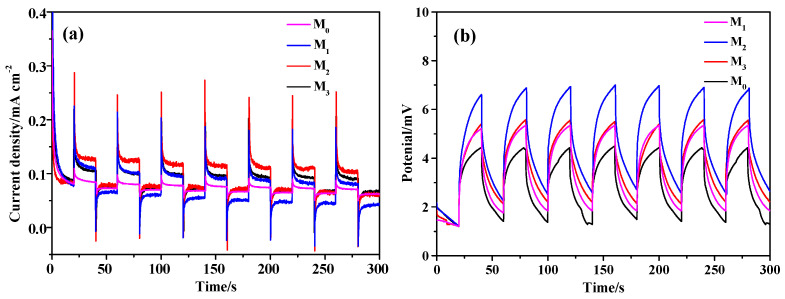
Photocurrent-time (**a**) and transient open-circuit voltage-time (**b**) curves of the prepared samples.

**Figure 8 molecules-28-07560-f008:**
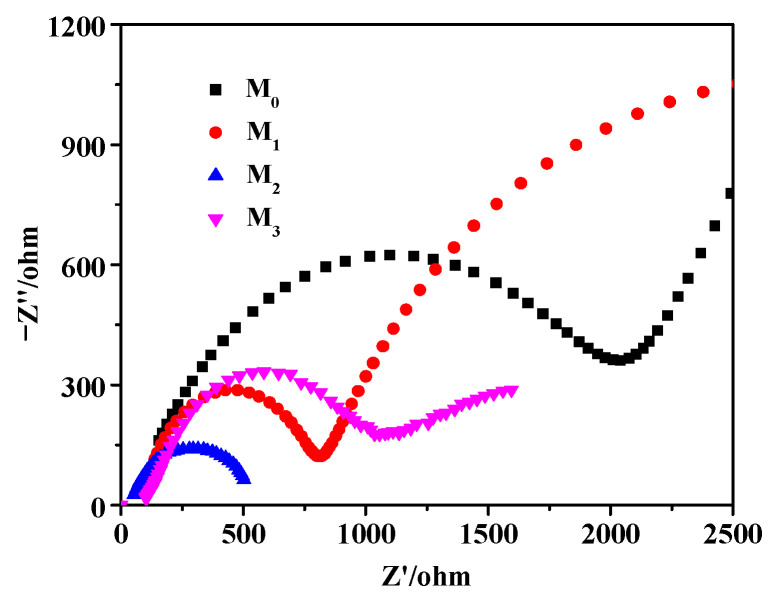
AC impedance spectra of the prepared samples.

**Figure 9 molecules-28-07560-f009:**
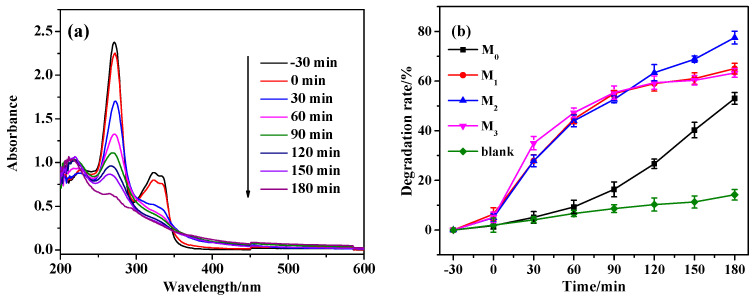
Degradation process (**a**) and degradation curve (**b**) of NOR.

**Figure 10 molecules-28-07560-f010:**
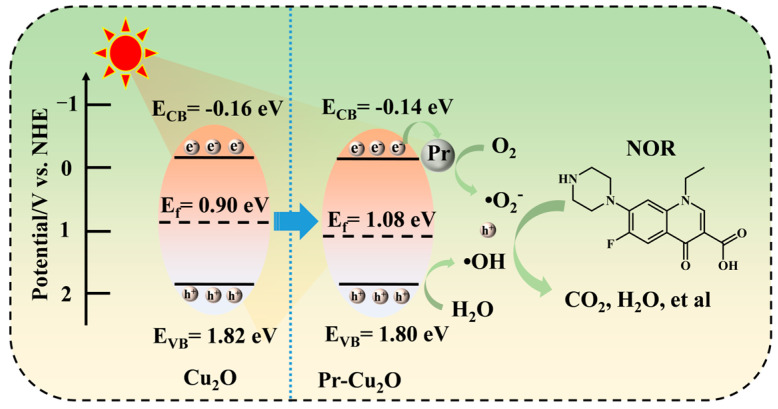
Schematic diagram of photocatalytic mechanism of Pr-Cu_2_O thin films.

**Figure 11 molecules-28-07560-f011:**
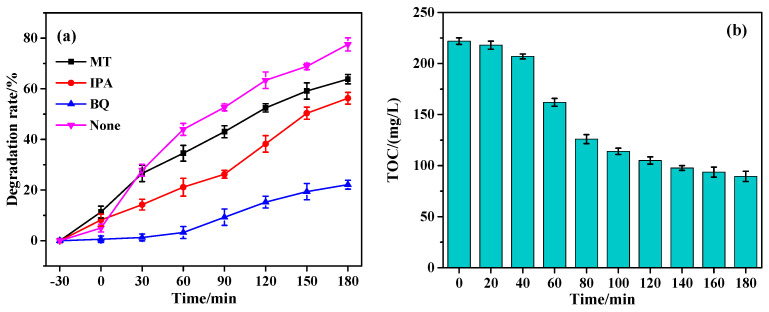
Effect of trapping agent on NOR degradation (**a**) and changes in TOC during degradation (**b**).

## Data Availability

Data are contained within the article.

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
