# Peer review of "Preparation and Photoelectric Properties of Pr-Doped p-Cu2O Thin Films Photocatalyst Based on Energy Band Structure Regulation"

_molecules, 2023, doi:10.3390/molecules28227560_

Round 1

Reviewer 1 Report

Comments and Suggestions for Authors

In this manuscript, the preparation and photoelectric properties of Pr-doped p-Cu2O thin films are presented. The effect of Pr doping to the structure, morphology and physicochemical properties of p-Cu2O was investigated. The authors evaluated the photocatalytic properties of the Pr-doped p-Cu2O thin films samples using norfloxacin as the target degradation material.

I recommend this paper for publication in Molecules.

The results are presented quite clearly, although some clarifications presented below would be necessary.

1) XRD: How does the crystallite size change after the introduction of Pr?

2) UV-Vis: What type of spectrophotometer was used for UV-Vis? It should be specified in the experimental part.

3) Since there is no evidence for the presence of Pr from the XRD and XPS measurements, the EDS analyzes (from the SEM) and the surface element composition resulting from the XPS tests should also be given, as well as the overall XPS spectrum. XPS measurements for the p-Cu2O sample (which does not contain Pr) should also be presented for comparison.

4) To analyze the degree of mineralization during the degradation process, total organic carbon (TOC) was determined. What method was used for this?

5) The authors present in the text that carbon dioxide or other products (which are not specified) result from the photocatalytic degradation reaction of norfloxacin. What method was used to measure the reaction products?

6) There are some typos in the text, for example:

-page 2, line 70: “The studies show that the the photoelectron…”;

-page 2, line 80: “It is expected that by introducing defects by doping. To change the band structure…”- There should be only one sentence here;

-page 2, line 84: “…and finally impeove the utilization…”-improve;

-page 3, line 105: “Therefore, deposition potential −4.5 V was selected for sample preparation.”- here it is actually −0.45 V;

- page 10, Figure 11 a): “Dragation rate”-Degradation rate;

-page 10, line 305: “…(ITO, square resistance ≤15Ω/â–¡)-≤15Ω/sq;

Author Response

Thank you for your letter on our paper entitled “Preparation and photoelectric properties of Pr-doped p-Cu2O thin films photocatalyst based on energy band structure regulation” (No.: molecules-2708475). We have carefully studied these opinions and made corrections, and the overall level of the paper has been greatly improved. We hope to receive approval. The major corrections in the paper are marked in red and the responses to the editor’s comments are as follows:

Responds to the editor’s comments:

Reviewer #1:

  1. XRD: How does the crystallite size change after the introduction of Pr?

Thank you for your valuable suggestion. The changes in grain size after the introduction of Pr element are described in lines 143-149 on page 4 of the revised manuscript, as follows: In addition, as the Pr doping amount increases from 0.9 mM to 1.5 mM, the grain size of the sample gradually decreases, but it is larger than that of the undoped Pr sample. This is because Pr doping can hinder the aggregation of Cu2O, increasing grain size. As the concentration of Pr increases, the agglomeration phenomenon gradually increases, which may be due to their extremely small size and high surface energy during crystallization and drying processes, resulting in a gradual decrease in size.

  1. UV-Vis: What type of spectrophotometer was used for UV-Vis? It should be specified in the experimental part.

In this article, UV-Vis was measured using UV-3600 (Shimadzu, Japan). Follow the suggestion, we have added this part in lines 325-326 on page 10.

  1. Since there is no evidence for the presence of Pr from the XRD and XPS measurements, the EDS analyzes (from the SEM) and the surface element composition resulting from the XPS tests should also be given, as well as the overall XPS spectrum. XPS measurements for the p-Cu2O sample (which does not contain Pr) should also be presented for comparison.

As mentioned by the reviewer, adding XPS and EDS analysis can help improve the persuasiveness of the article. As we described, the binding energy of Pr 3d was too close to that of Cu 2p orbitals and the content was low, so it was not detected in the XPS full spectrum (Figure 3c). We confirmed the presence of the Pr element in the M2 sample through EDS analysis, but its content was only 0.11% (Figure 3d), consistent with XPS analysis. We added this section on page 4, lines 127-129, and the relevant images are Figures 3c and 3d.

  1. To analyze the degree of mineralization during the degradation process, total organic carbon (TOC) was determined. What method was used for this?

Due to our negligence, there was no explanation regarding the total organic carbon (TOC) detection. In this article, we use an elemental vario TOC (Germany) instrument to detect the mineralization of the degradation solution. We added this section on page 10, lines 326-328.

  1. The authors present in the text that carbon dioxide or other products (which are not specified) result from the photocatalytic degradation reaction of norfloxacin. What method was used to measure the reaction products?

In this article, we use a UV-Vis spectrophotometer (UV-3200PCS, Shanghai Meipuda Instrument Co., Ltd.) to monitor the concentration changes of NOR. We added this section on page 10, lines 328-330. Regarding the decomposition of NOR into CO2 and H2O, this is the most ideal decomposition product inferred based on the production of TOC.

  1. There are some typos in the text, for example:

-page 2, line 70: “The studies show that the the photoelectron…”;

-page 2, line 80: “It is expected that by introducing defects by doping. To change the band structure…”- There should be only one sentence here;

-page 2, line 84: “…and finally impeove the utilization…”-improve;

-page 3, line 105: “Therefore, deposition potential −4.5 V was selected for sample preparation.”- here it is actually −0.45 V;

- page 10, Figure 11 a): “Dragation rate”-Degradation rate;

-page 10, line 305: “…(ITO, square resistance 15Ω/â–¡)-15Ω/sq;

We deeply apologize for any typos in the text. We have corrected all errors listed by the reviewers, and in addition, we have checked the entire text to correct spelling errors.

Reviewer 2 Report

Comments and Suggestions for Authors

1. The introduction does not discuss other photocatalysts and the benefits of Cu2O

2. Not all statements in the introduction are supported by literature sources

3. The description of characterization method need to be improved. For example, how the calibration of the binding energy scale was performed in XPS technique? What analyzer was used in XPS? What detector was used in XRD?

4. In Fig 11, please add experimental errors 

5. Please add calculations of the absolute electronegativity parameter

6. Please add scheme of the photocatalytic reaction instrument. What is the concentration of photocatalyst in the experiments?

Comments on the Quality of English Language

There are typos in the work, for example:

impeove

It is expected that by introducing defects by doping. To change the band structure of Cu2O, ...

Author Response

Thank you for your letter on our paper entitled “Preparation and photoelectric properties of Pr-doped p-Cu2O thin films photocatalyst based on energy band structure regulation” (No.: molecules-2708475). We have carefully studied these opinions and made corrections, and the overall level of the paper has been greatly improved. We hope to receive approval. The major corrections in the paper are marked in red and the responses to the editor’s comments are as follows:

Responds to the editor’s comments:

Reviewer #2:

  1. The introduction does not discuss other photocatalysts and the benefits of Cu2

Thank you for your question. Regarding the description of other catalysts and Cu2O, we have added lines 44 to 51 on page 2. The specific content is as follows: Photocatalyst is the core of photocatalytic oxidation technology, in which the semiconductor photocatalyst has become the focus of research in recent years because it can use sunlight as a driving force to degrade organic pollutants. Therefore, it is crucial to explore an ideal photocatalyst with high light absorption, excellent photo-catalytic performance, and outstanding stability (e.g., BiOBr, WO3, and BiPO4). As a typical p-type semiconductor, Cu2O is widely used for pollutant photodegradation, nitrogen fixation, and carbon dioxide reduction due to its suitable bandgap, strong response to visible light, low cost, and non-toxic properties.

  1. Not all statements in the introduction are supported by literature sources

Thank you for your question. Due to our negligence, the statements in the introduction lacked literature support, so we added corresponding literature in the description section of the introduction and marked it in red. References 2-8, 12, and 13 have been added.

  1. The description of characterization method need to be improved. For example, how the calibration of the binding energy scale was performed in XPS technique? What analyzer was used in XPS? What detector was used in XRD?

Due to our negligence, the description of the characterization method needs improvement. XPS uses the C 1s transition at 284.6 eV as an internal reference to adjust the binding energy scale of the peaks obtained in the experiments conducted. XRD detection is performed using the XRD-6100 instrument (λ=1.54056 Å, Shimadzu Instrument Co., LTD., Japan), Cu Target K α Radiation. We have added lines 320 to 331 on page 10.

  1. In Fig 11, please add experimental errors

As the reviewer said, adding error bars can make the data more convincing. We have added error analysis in both Figure 9 (page 8, line 227) and Figure 11 (page 10, line 304).

  1. Please add calculations of the absolute electronegativity parameter.

Thank you for your correction. We have added the calculation formula for absolute electronegativity on page 8, lines 243-246.

  1. Please add scheme of the photocatalytic reaction instrument. What is the concentration of photocatalyst in the experiments?

In this article, the concentration of the catalyst is 400 mg·L-1. We will add this section on page 10, line 334.

  1. There are typos in the work, for example:

impeove

It is expected that by introducing defects by doping. To change the band structure of Cu2O, ...

We deeply apologize for any spelling errors in the text. We have corrected all spelling errors raised by the author, and in addition, we have checked the spelling issues throughout the entire text.

Round 2

Reviewer 2 Report

Comments and Suggestions for Authors

thanks for correction, the arcticle is recommnended for publication